# [Re] Assessing the Reliability of
# Word Embedding Gender Bias Measures

## Reproducibility Summary

**Scope of Reproducibility**

This work attempts to reproduce the results of the paper *Assessing the Reliability of Word Embedding Gender Bias Measures* by Du et al. (2021). In this paper, the authors test to which extent gender bias measures are consistent and reliable in popular word embedding models. The main claims of the original paper with regard to word embedding gender bias measures are:

1. High test-retest reliability
2. High internal consistency
3. Low inter-rater consistency

It is important to verify results claimed by studies, in order to preserve the integrity of scientific research. Therefore, the scientific community has encouraged reproducing papers in order to make researchers more aware about the reproducibility of their future work. We support this movement by contributing through reproducing the work done by Du et al. (2021).

**Methodology**

We used the author's code to attempt reproducing the results. Furthermore we investigated whether the evaluation framework proposed by the authors would also be applicable to other forms of bias. Therefore we altered the code to assess the reliability of measuring sexual orientation bias in word embeddings. The experiments were run on a machine with a Intel i7-8700 CPU and a machine with a Bronze 3104 CPU. The total running time (sequentially) was roughly 150 hours.

**Results**

The reproduced results mainly agree with the claims made by the original paper. However, we found that the variance of the test-retest reliability scores depends on the batch of random seeds used. Therefore we suggest that more random seeds are needed to support the first claim made by the original paper, which is high test-retest reliability.

**What was easy**

The paper was very well-documented. It was clear what they wanted to test, how they tested it, and what conclusions they could draw from it.

**What was difficult**

However, this clear documentation of the paper was not entirely reflected in their code. There were many bugs in the code and the README file did not contain all the information to successfully reproduce the experiments.

# 1 Introduction

Natural language processing (NLP) techniques play an important role in our life. This is why NLP systems need to be fair, socially responsible and accountable. Many research groups devoted their studies to ensure fairness across relevant topics within today's zeitgeist, one of these topics is gender bias in embedding models. Multiple measures of bias have been proposed and are now widely used (Bolukbasi et al., 2016; Caliskan et al., 2017; Ethayarajh et al., 2019; Gonen and Goldberg, 2019). The paper which we will assess in this work, *Assessing the Reliability of Word Embedding Gender Bias Measures* by Du et al. (2021), tests to which extent such gender bias measures are consistent and reliable.

In this work we assess the assessment done in the original paper by attempting to reproduce their results, using the code provided by the authors. The topic of reproducibility has become widely discussed in the scientific community. It is important to verify results claimed by studies, in order to preserve the integrity of scientific research. However, there are estimates that as much as 50% of published studies, even those in top-tier academic journals, cannot be repeated with the same conclusions by an industrial lab (Crick et al., 2017; Osherovich, 2011; Saey, 2015). Therefore, the scientific community has encouraged reproducing papers in order to make researchers more aware about the reproducibility of their future work. We support this movement by contributing through reproducing the work done by Du et al. (2021).

# 2 Scope of reproducibility

The main theme of the work by Du et al. (2021) is assessing whether gender bias measures of word embedding models produce consistent results to a certain extent.

Gender bias measures work more or less in the following way (Du et al., 2021): to calculate the gender bias score of a target word, the word embedding of this target word $w$ is taken into account. Additionally, the word embeddings $m$ and $f$ of a male-female gender base pair are constructed. A male-female gender base pair is a pair of words which are obvious gender equivalent synonyms with respect to each other, e.g. *man - woman* or *king - queen*. Then there is a scoring rule, which is a function that takes the word embeddings $w$, $m$ and $f$ as input and outputs a bias score. These bias scores are primarily based on similarity or distance between the target word embedding $w$ and the gender specific word embeddings $m$ and $f$. It is regular to use multiple gender base pairs, after which an average is calculated.

It is evident that the results are dependent on the choice of target words, gender base pairs and scoring rules, which is the motivation for the work done by Du et al. (2021). In their paper they assess whether there is consistency in producing results among using different random seeds (*test-retest reliability*), different scoring rules (*inter-rater consistency*) and different target words (*internal consistency*).

The main claims found by the original paper after assessment were the following:

1. *High test-retest reliability:* bias scores are mostly consistent across different random seeds. The consistency of bias scores across different random seeds are mostly influenced by various word-level features as well as the word embedding algorithm used.

2. *High internal consistency:* bias scores are mostly consistent across gender base pairs and target words within a query.

3. *Low inter-rater consistency:* the three scoring rules fail to agree with one another. Bias scores of target words across different scoring rules are dominated by the word embedding.

We will test whether the three claims provided by the original paper are consistent by reproducing their work.

# 3 Methodology

We used the code provided by the authors in their Github repository. However, not all code was provided: in order to train the GloVe embeddings we had to copy the code from the GloVe repository and adjust it to the task at hand. The final results, consisting of figures, were all created from within a Jupyter Notebook. All required steps for reproduction are documented in the README files. The provided Python code from GitHub, including the Jupyter Notebook, were adjusted to enable the usage of a different base pair list. The original and adjusted code is not optimized for GPU usage.

## 3.1 Model descriptions

The paper assessed two popular unsupervised embedding models: Skip-Gram with Negative Sampling (SGNS) and Global Vectors (GloVe). Both models vectorize textual data into so-called word embeddings.

Skip-Gram is a relatively simple neural network with one hidden layer without activation function and an output layer with a SoftMax classifier (Mikolov et al., 2013a). For each word, a one-hot encoded vector of the vocabulary size is constructed. Then the zero's are replaced by the probability that the corresponding word is in proximity to the target word, which is ensured by the SoftMax function. This will result in higher probabilities for words that often appear together. In order to decrease computational time for training, Mikolov et al. (2013b) introduced Negative Sampling, which is a simple training method that learns accurate representations especially for frequent words.

Where Skip-Gram captures local statistics, GloVe builds word embeddings on global statistics (Pennington et al., 2014). A co-occurence matrix is constructed for all words within the vocabulary. This matrix is then used to construct word embeddings .

We trained the embedding models on each of the corpora (which will be mentioned in section 3.2) separately. This was repeated 32 times, using different random seeds in order to measure test-retest reliability.

## 3.2 Datasets

We used three different corpora to train the embedding models on: the comments from two subReddits from 2014, *r/AskScience* with ~158 million tokens, *r/AskHistorians* with ~137 million tokens, and lastly, the training set of WikiText-103 with ~527 million tokens, consisting of high-quality Wikipedia articles. You can download the Reddit corpora by the following links: `https://files.pushshift.io/reddit/comments/`

Download all the files of reddit comments in 2014. Or the code provided by Du et al. (2021) can be used which contains a script to subtract the subreddits. For WikiText-103 the following link can be used: `https://s3.amazonaws.com/research.metamind.io/wikitext/wikitext-103-v1.zip`

Furthermore, additional word lists are used to assess test-retest reliability and inter-rater consistency. Three of these word lists are specific for gender bias and used in previous studies, they include 320 occupation words by Bolukbasi et al. (2016), 76 additional occupation words by Garg et al. (2018), and 230 adjectives by Garg et al. (2018). Larger, more generic word lists are also included so this can also be applied to future work about other forms of bias. These are Google10K (the 10.000 most frequent words of Google Web Trillion Word Corpus) and the full vocabulary of each corpus. It can be found here: `https://github.com/first20hours/google-10000-english`

## 3.3 Hyperparameters

For the both models we used the default parameters, except for the vector size, or embedding dimension, which was set to 300 to match the vector size used by Du et al. (2021).

## 3.4 Experimental setup and code

In order to investigate test-retest reliability, internal consistency and inter-rater consistency of gender bias measures, Du et al. (2021) proposed three different experiments.

For examining test-retest reliability, gender bias scores of word embedding models (for all target word lists, scoring rules and gender base pairs) were calculated 32 times, using different random seeds.

Inter-rater consistency was examined by comparing the bias scores using different scoring rules. DB/WA (Direct Bias / Word Association), RIPA (Relational Inner Product Association) and NBM (Neighbourhood Bias Metric). They approached this with two different schemes: 1. assessing the bias scores of a single target word averaged over all gender base pairs, and 2. assessing the bias scores using a single gender base pair, averaging over all target words.

Lastly, for the examination of internal consistency also two schemes were investigated: 1. to test the consistency of a query (a subset of related target words), the bias scores across different target words within a query were compared, and 2. to test the internal consistency of gender base pairs, the bias scores of different gender base pair, averaged over all target words, were compared.

The authors made an open source Github repository of their code with a corresponding README file. One can find it here: `https://github.com/nlpsoc/reliability_bias`

We reproduced the paper by running the code ourselves. However, the code for the GloVe embedding model was not provided but had to be retrieved from a Github repository. This code was used to train the model with only a single adjustment. The vector size was changed to 300 in *demo.sh*. One can find the Github repository here: `https://github.com/stanfordnlp/GloVe`

During reproducing the paper, the question arose whether the evaluation framework proposed by the authors would also be applicable to other forms of bias. Therefore all experiments were executed again, except this time with base pairs concerning sexual orientation. The intention is to assess the reliability of measuring bias in terms of the 'conventional' sexual orientation, which society depicts as heterosexualism against all other 'non-conventional' sexual orientations. The used sexual orientation base pairs are displayed in appendix A. The target word lists which were selected for gender bias, e.g. occupations and adjectives, form similarly interesting target words for sexual orientation bias. Therefore the target word lists were kept unchanged.

Our final code can be found here: `https://anonymous.4open.science/r/MLRC-2021-CD10/`

## 3.5 Computational requirements

In this research two different machines were used for the computations. The relevant details for both machines are displayed in table 1.

The Glove model was trained on machine 1. A GPU was in fact available but the code provided was not optimized for GPU calculations. The training time for a single embedding model differs per corpora, 22 minutes for r/AskScience, 28 minutes for r/Askhistorians and 150 minutes for WikiText-103. As 32 different random seeds were used the total training time for all the Glove embedding models was 107 hours. To estimate the training time for a single model for a new corpora one can best look at the size of the co-occurrence file, as it has the highest linear correlation. A co-occurrence file of 100mb corresponds roughly with 4 minutes training time.

The skip Gram model was trained on machine 2. The training of a single Skip-Gram with Negative Sampling embedding took 4 minutes for the WikiText-103 corpus, 1 minute for r/AskScience and also 1 minute for r/AskHistorians. The total training time is approximately 3 hours. The calculating of bias scores for a a single Skip-Gram with Negative Sampling embedding took 4 minutes for the WikiText-103 corpus and less than a minute for both r/AskScience and r/AskHistorians. The total calculation time is about 2 hours. Together with the training of the embeddings, the total required CPU hours is 5 hours. The GloVe model was not tried on Machine 2 due to an expected RAM and storage shortage.

|  | Processor | CPU Cores | RAM Memory |
|---|---|---|---|
| Machine 1 | Bronze 3104 (1.7GHz) | 3 | 64GB |
| Machine 2 | Intel i7-8700 (3.2GHz) | 6 | 16GB |

Table 1: Relevant details about the machines used in this research

# 4 Results

## 4.1 Results reproducing original paper

### 4.1.1 Test-Retest Reliability

The distributions of the test-retest reliability scores across target word lists and scoring rules (based on SGNS and WikiText-103) are displayed in figure 1. The left plots show the reproduced results whereas the right plots are the results from the original paper. The reproduced scores seem to agree with the original scores in terms of their absolute values and their relative values with respect to each other. However, there is a significant difference in the interquartile ranges and overall ranges. The test-retest reliability scores of the target words and gender base pairs of the reproduced results have lower variance than their original counterpart.

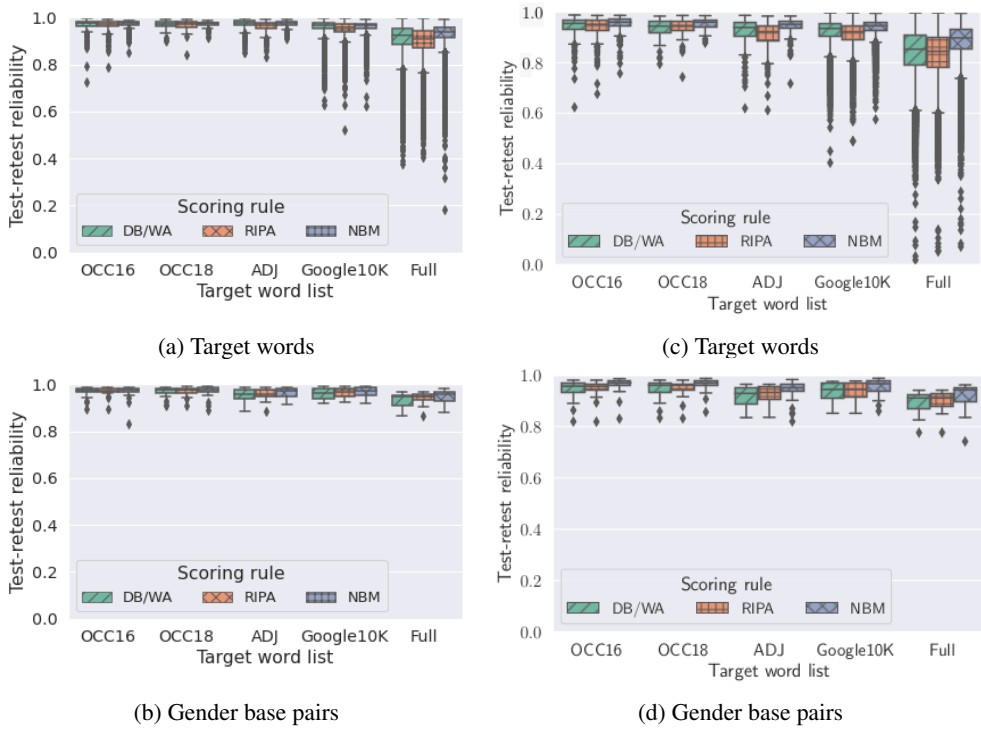

(a) Target words

(c) Target words

(b) Gender base pairs

(d) Gender base pairs

Figure 1: Distribution of the test-retest reliability scores across target word lists and scoring rules (based on SGNS and WikiText-103). Plots a and b are the reproduced results and plots c and d are the results from Du et al. (2021).

### 4.1.2 Inter-rater Consistency

When comparing the reproduced results with the original results as can be seen in figure 2, a similar observation Du et al. (2021) can be made. The inter-rater consistency for the majority of both the target words as the gender base pairs is low. For the target words plot (a), the r/AskHistorians corpus performs better than the other two corpora while for the gender base pairs the corpora perform similar. More similar results for other combinations of corpora and embedding models can be found in the complete list of results in appendix C under section 'Inter-Rater Consistency' part (a) regarding gender base pairs.

For further analysing the similarities between the bias scores of the target words, the bias scores for all corpora are combined and their Pearson similarity is calculated. For the bias scores calculated on the GloVe embeddings, the least similarity is between DB/WA and NBM (Pearson's $r$: 0.810, $p < 0.05$). The comparison between RIPA and NBM gives a similar result (Pearson's $r$: 0.816, $p < 0.05$). The bias scores of DB/WA and RIPA are the most similar (Pearson's $r$: 0.989, $p < 0.05$). These results together with the results from the original paper are shown in table 2. More similarity scores can be found in appendix C under section 'Inter-Rater Consistency' part (c).

|  | Reproduced | Du et al. (2021) |
|---|---|---|
| RIPA & NBM | 0.816 | 0.836 |
| DB/WA & RIPA | 0.989 | 0.923 |
| DB/WA & NBM | 0.810 | 0.897 |

Table 2: Comparison of Du et al. (2021) and the reproduced Pearson's $r$ scores for bias scores of all corpora trained on GloVe embeddings. All with significance $p < 0.05$.

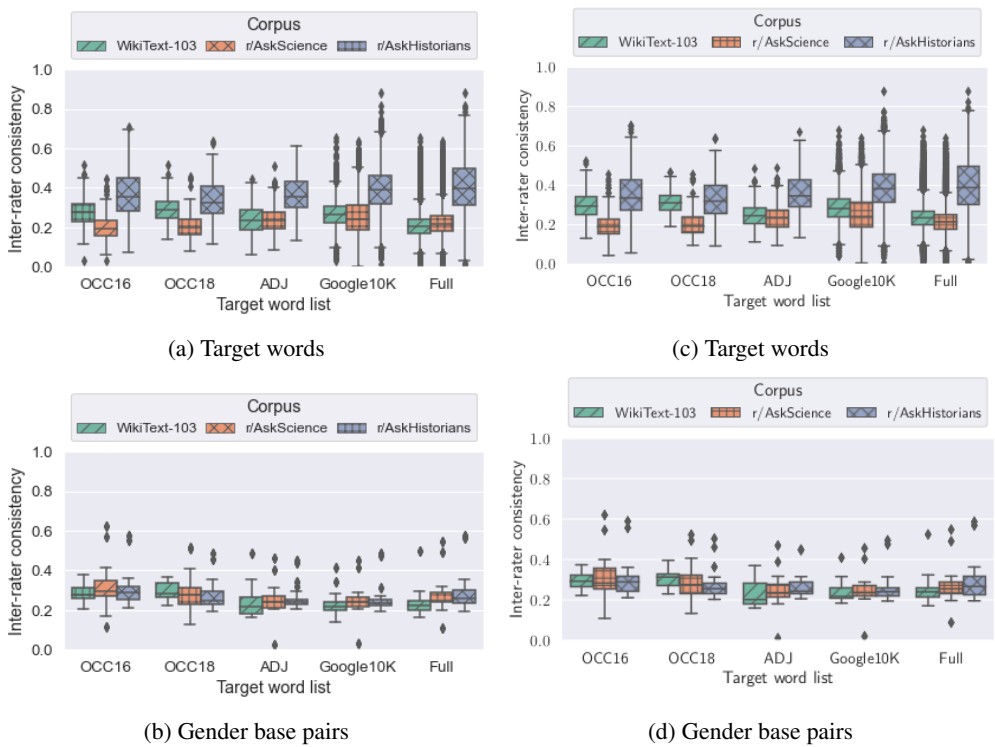

(a) Target words

(b) Gender base pairs

(c) Target words

(d) Gender base pairs

Figure 2: Distribution of inter-rater consistency scores across target word lists and corpora (based on GloVe). Plots a and b are the reproduced results and plots c and d are from Du et al. (2021).

### 4.1.3 Internal Consistency

In general the reproduced results for the internal consistency agree with the results produced by Du et al. (2021), see figure 3. Minor differences in the ranges and whiskers can be seen but all the ranges are encompassed by the whiskers of both the reproduced and the original work.

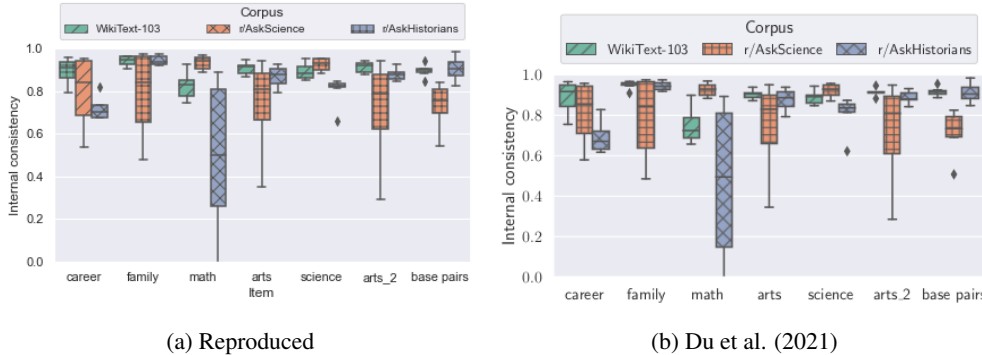

(a) Reproduced

(b) Du et al. (2021)

Figure 3: Distribution of internal consistency scores of gender bias related queries (e.g., career) and the ensemble of gender base pairs (base pairs) across corpora. Each query and the ensemble of gender base pairs has six reliability scores across different combinations of embedding algorithms and scoring rules.

### 4.1.4 Factors influencing the reliability of gender base pairs and target words

The visualizations of the inter-rater consistency and test-retest reliability scores per gender base pair can be seen in figures 39 and 38 in appendix C. The reproduced test-retest reliability scores seem to have higher whiskers than the

original scores, although the magnitudes are similar. The reproduced inter-rater consistency scores are similar to the original ones.

The results of multilevel regression on the test-retest and inter-rater reliability of target words can be found in table 3 in appendix C. Complete reproduction of the original paper was not possible due to various reasons, further explanation and description can be found in the discussion.

## 4.2 Testing whether the framework is also applicable to sexual orientation bias

For the test-retest reliability when using sexual orientation base pairs, the interquartile range is for a majority of the plots larger when compared with the interquartile ranges of gender base pairs. Furthermore, the median values are mostly equal but in several cases higher, which can clearly be seen when comparing figure 9 and 10 with figure 21 and 22 from appendix C.

The inter-rater Consistency plots for sexual orientation base pairs show comparable results to the gender base pair plots. With equally comparable Pearson similarity scores for the scoring rules which can be seen by comparing tables 4 and 5. The internal consistency for sexual orientation, figure 37, again shows a larger range and larger interquartile ranges for all queries. The sexual orientation base pairs internal consistency (rightmost box) is significantly lower than the gender base pairs internal consistency.

The reliability analysis for sexual orientation base pairs (figures (41 and 40) shows very similar results to the gender base pairs reliability analysis (figures 39 and 38). These tables and figures can be found in appendix C.

## 5 Discussion

### 5.1 Test-Retest Reliability

It is evident from figure 1 that although the magnitudes of the test-retest reliability scores of the reproduced results agree with the original results, both in absolute value and their internal differences, the variance of the scores is significantly lower compared to the original scores. As the only difference between the reproduced and original results is the random seeds, this may suggest that 32 random seeds are not enough to produce consistent results. However, this does not necessarily reject the first claim of the paper, which is high test-retest reliability. Because of the lower variance of the results in combination with the same high absolute values, this even supports the claim that bias scores are independent among different random seeds. Nonetheless it should be noted that with the use of another batch of 32 random seeds, the variance may turn out to be higher as we already deduced that the variance is dependent on the random seeds. Therefore we suggest that the use of more random seeds should be investigated in order to support the claim.

### 5.2 Inter-Rater Consistency

The low inter-rater Consistency scores for both the target words as well as the gender base pairs in figure 2, indicate that the three scoring rules RIPA, NBM and DB/WB provide very different bias scores and may be measuring different aspects of word embedding gender biases. Which is in accordance with the observations of Du et al. (2021). Explaining the better scores for the target words on the r/AskHistorians corpus would require further analysis of comparing specific bias scores of the target words. This would make interesting future work but will not be performed in this reproducability paper.

The Pearson's similarity scores from table 2 show that DB/WA and NBM do not necessarily deliver more similar bias scores than RIPA and NBM, despite being based on scores of the nearest neighbours and RIPA and NBM not sharing a comparable calculation method. The bias scores between DB/WA and RIPA are very similar, which could be explained by their usage of cosine similarity as stated in Du et al. (2021).

### 5.3 Internal Consistency

From figure 3 we can deduce there are some minor differences between the reproduced and original internal consistency scores: the whiskers and variance of the reproduced results are slightly smaller. However, the magnitudes of the reproduced scores are similar to the original ones, which according to Du et al. (2021), are relatively high. This combination of small whiskers and variance together with high magnitudes of the reproduced internal consistency scores supports the claim made in the paper, which is high internal consistency.

### 5.4 Factors influencing the reliability of gender base pairs and target words

The visualizations of the inter-rater consistency and test-retest reliability scores per gender base pair can be seen in figures 39 and 38 in appendix C. Just like with the reproduced results of test-retest reliability from section 4.1.1, the magnitudes of the scores agree with the results of the original paper, but the whiskers are different. As such, we come to the same conclusion and suggest to use more than 32 random seeds in order to produce consistent results among test-retest reliability. The visualization of the inter-rater consistency score is similar to the one displayed in the original paper. However, in the original paper this visualization is compared to the visualization of test-retest reliability per gender base pair. As we already concluded that more random seeds are needed to provide consistent results, we cannot verify their comparisons.

The results of multilevel regression on the test-retest and inter-rater reliability of target words can be found in table 3 in appendix C. The results are incomplete due to various reasons. Firstly, the provided code did not work for all corpora and there was no solution found. Secondly, various parts in the code reference a certain dispersion variable which is not defined and causes the code to crash, therefore it was removed. Finally the final results for the multilevel regression code are ambiguous, as for example there is no mention of any R-squared values. The inconsistency in coding language (R was used instead of Python) and a lack of useful comments, prevented us from debugging further. Although the rest of the code is reproducible to some extend, this part does not yield any valid results.

### 5.5 Testing whether the framework is also applicable to sexual orientation bias

The provided framework is suited for replicating results with a different set of base pairs. The original claims of Du et al. (2021), high test-retest reliability, low inter-rater consistency and high internal consistency apply to the results of using sexual orientation base pairs. However, the internal consistency of sexual orientation base pairs contains for the queries, more variance and slightly lower medians (figure 37 in appendix C). A possible explanation could be the lower frequency of the sexual orientation base pairs in the corpora. The target words are found in less different context which could increase the variance and overall lower consistency. For the base pairs, the Cronbach's alpha is smaller than 0.7 for all three corpora which is below acceptable (Cicchetti, 1994). This indicates that the bias scores strongly differ between sexual orientation base pairs. It could therefore be the case that the manually selected sexual orientation base pairs do not cover the the same concept of sexual orientation bias.

### 5.6 What was easy

The paper was very well-documented and had an easy-to-follow structure. It was clear what the authors wanted to test, how they tested it, and what conclusions they could draw from it. As the topic of the paper was reliability and consistency, which are important themes within the scope of reproducibility, the authors probably kept in mind that writing a clear paper is crucial to make reproducibility possible.

### 5.7 What was difficult

However, this clear documentation of the paper was not entirely reflected in their code. First of all the README file had minimal information and the comments - or lack of it - were generally not clear. Some parts of the code were also buggy, incomplete or even completely missing. Altogether only trying to run the provided code was a trysome experience which took a lot of time. More specific points of critique are described below, and a full list can be found in Appendix B.
The Jupyter Notebook reliability_analyses.ipynb was created with the intention of producing results for the gender base pairs. Several large modification had to be made to the notebook and the related files to enable the usage of a different base pair list, the sexual orientation base pairs.
Some crucial information to ensure successful reproducibility was missing in the README, namely, setting the vector size of the GloVe model to 300. They did state this in the paper however. Nonetheless, we think that such code-related details should be stated in the README. Several alterations had to be made to the code to be able to run, specifics can be found in Appendix B.

### 5.8 Communication with original authors

There has been no communication with the original authors.

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

# Appendix A

The base pairs we used to test whether the evaluation framework proposed by the authors of the original paper could also be applied to sexual orientation bias are displayed below.

"straight" - "gay"

"straight" - "lesbian"

"straight" - "bi"

"heterosexual" - "homosexual"

307  "heterosexual" - "bisexual"

308  "heterosexual" - "asexual"

309  "heterosexual" - "pansexual"

310  "hetero" - "homo"

311  "heterosexualism" - "homosexualism"

312  "heterosexualism" - "bisexualism"

313  "heterosexualism" - "asexualism"

314  "heterosexualism" - "pansexualism"

# Appendix B

In this section some additional bugs or incomplete code are described.

- In  ./mlr/calc_word_property.py  the  following  line  should  be  added  after  line  125: glove_embed_models.append(embed_model).  As this lack of an append becomes only relevant at the end of the code the code crashed after two hours of running.

- In ./mlr/calc_word_property.py several nltk corpus are used and imported. Some code should be added which allows the user to download the corpus' if they have not already.

- In ./mlr/multilevel_test_retest.R and ./mlr/multilevel_inter_rater.R all the lines mentioning dispersion were removed. We are under the impression that the authors forgot to remove this part completely.

- All the files that are opened in the code should be opened with the help of the library IO. Furthermore without specifying the encoding to be uft8 the code would not run. This may vary per machine.

- In train_sgns,py several arguments of Word2Vec() should be changed to their updated variant:

  – 'size' $\implies$ 'vector_size'

  – 'iter' $\implies$ 'epochs'

- In data_loader.py a deprecated call is given to the loaded Word2Vec gensim model in the load_gensim_sgns() function. A change should be made:

  – 'embed_model.vocab' $\implies$ 'embed_model.key_to_index'

- Not all required folders are provided in the GitHub repository. However they are specified in the paths.py folder. Therefore extra code is added to paths.py to create the required folders automatically.

- It is advised to use a Linux OS when running the code, however when using Windows, the print() function which is used to write to a file in ./data/train_corpora/reddit/get_subreddit.py may lead to encoding problems. A different writing method such as file.write() did not lead to an ecoding error. This may vary per machine.

- The ./data/train_corpora/reddit/reddit_dl.sh file contains url's to https://archive.org/ for downloading the subreddit comments. The download speed from this source is very restricted. As stated in section 3.2, a different source could be used such as http://files.pushshift.io/reddit/comments delivering significantly faster download speeds.

 **Appendix C**

| | Test-retest | | Inter-rater | |
|---|---|---|---|---|
| | Estimate | $\Delta R^2$ | Estimate | $\Delta R^2$ |
| $SR_{DB/WA}$ | reference | | - | - |
| $SR_{RIPA}$ | -0.0109 | | - | - |
| $SR_{NBM}$ | 0.0052 | | - | - |
| log freq | 0.0202 | | 0.0072 | |
| $log^2 freq$ | -0.0158 | | 0.0064 | |
| log #senses | 0.0008 | | -0.0005 | |
| $POS_{adj}$ | reference | | reference | |
| $PoS_{adp}$ | 0.0026 | | -0.0146 | |
| $PoS_{adv}$ | 0.0048 | | 0.0047 | |
| $PoS_{conj}$ | -0.0591 | | -0.0113 | |
| $PoS_{det}$ | 0.0048 | | 0.0012 | |
| $PoS_{noun}$ | -0.0021 | | 0.0071 | |
| $PoS_{num}$ | 0.0077 | | 0.0038 | |
| $PoS_{pron}$ | -0.0008 | | 0.0315 | |
| $PoS_{prt}$ | 0.0096 | | -0.0063 | |
| $PoS_{verb}$ | -0.0023 | | 0.0022 | |
| $PoS_x$ | -0.0036 | | -0.0100 | |
| NN Sim | -0.0111 | | -0.0073 | |
| L2 norm | -0.0291 | | -0.0173 | |
| ES | 0.1315 | | 0.0810 | |
| R^2_{fixed} | | | | |
| R^2_{corpus} | | | | |
| R^2_{algorithm} | | | | |
| R^2_{total} | | | | |

Table 3: Results of multilevel regression on the test-retest and inter-rater reliability of target words. $\Delta R^2$ is reduction in explained variance when the corresponding factor is left out. $R^2$ fixed, $R^2$ corpus, $R^2$ algorithm and $R^2$ total refer to the explained variance of fixed factors (i.e. word level features and scoring rules), embedding training corpus used, embedding training algorithm used, total effects of all these three parts, respectively

 The additional plots of the results are displayed below.

343 **Test-Retest reliability**

344 **(a) Gender base pairs**

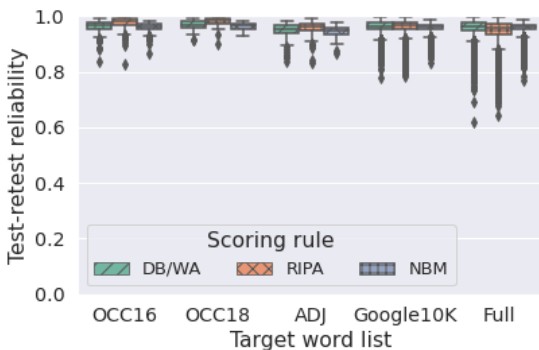

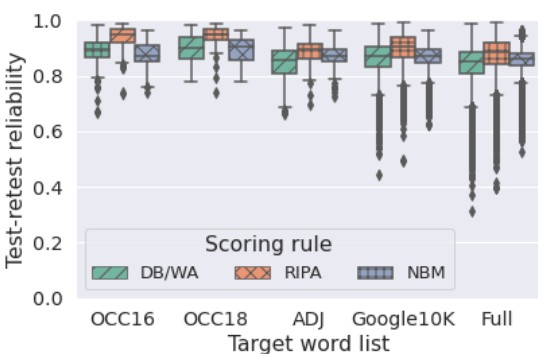

Figure 4: Test-retest reliability of target words (bias scores averaged across gender base pairs). The word embeddings are trained with SGNS on r/AskHistorians.

Figure 5: Test-retest reliability of target words (bias scores averaged across gender base pairs). The word embeddings are trained with SGNS on r/AskScience.

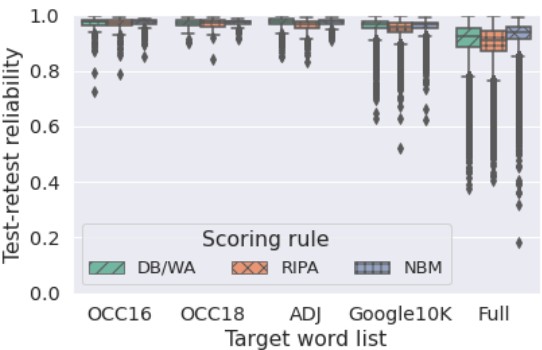

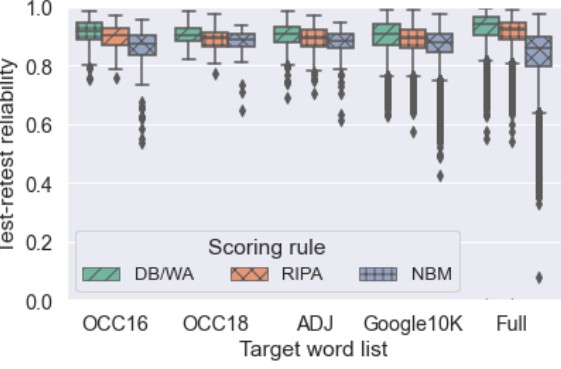

Figure 6: Test-retest reliability of target words (bias scores averaged across gender base pairs). The word embeddings are trained with SGNS on WikiText-103.

Figure 7: Test-retest reliability of target words (bias scores averaged across gender base pairs). The word embeddings are trained with GloVe on r/AskHistorians.

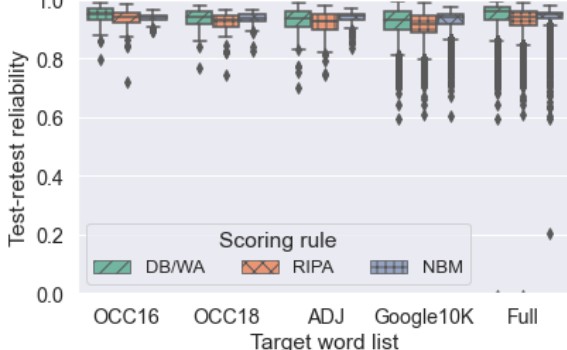

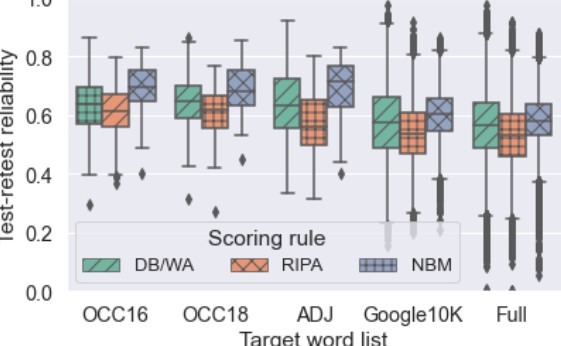

Figure 8: Test-retest reliability of target words (bias scores averaged across gender base pairs). The word embeddings are trained with GloVe on r/AskScience.

Figure 9: Test-retest reliability of target words (bias scores averaged across gender base pairs). The word embeddings are trained with GloVe on WikiText-103.

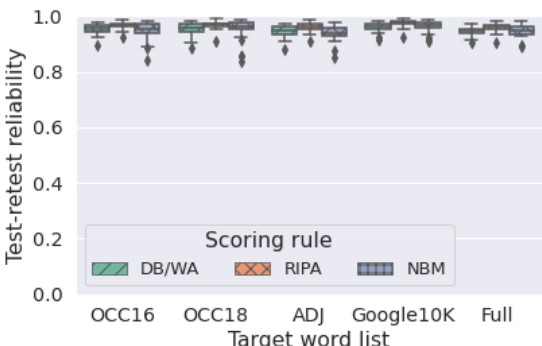

Figure 10: Test-retest reliability gender base pairs. The word embeddings are trained with SGNS on r/AskHistorians.

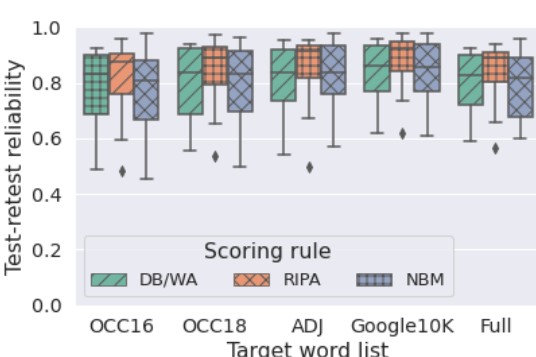

Figure 11: Test-retest reliability gender base pairs. The word embeddings are trained with SGNS on r/AskScience.

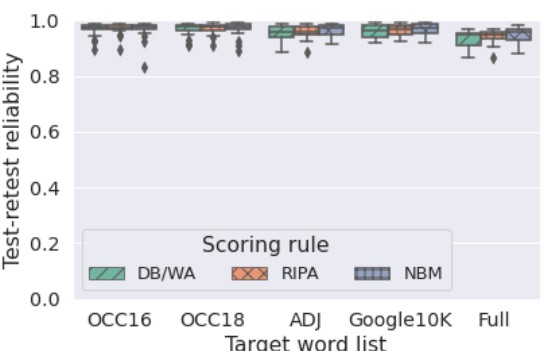

Figure 12: Test-retest reliability gender base pairs. The word embeddings are trained with SGNS on WikiText-103.

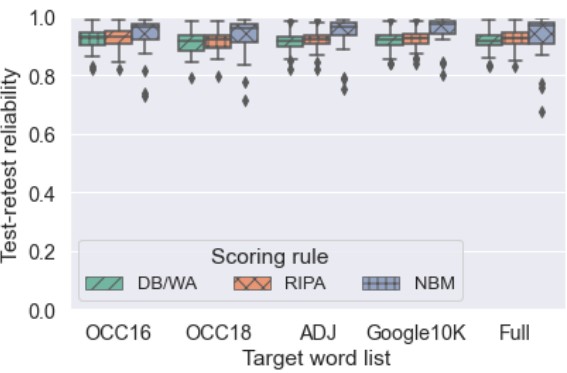

Figure 13: Test-retest reliability gender base pairs. The word embeddings are trained with GloVe on r/AskHistorians.

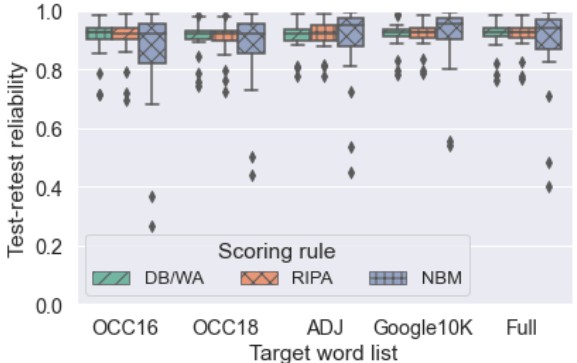

Figure 14: Test-retest reliability gender base pairs. The word embeddings are trained with GloVe on r/AskScience.

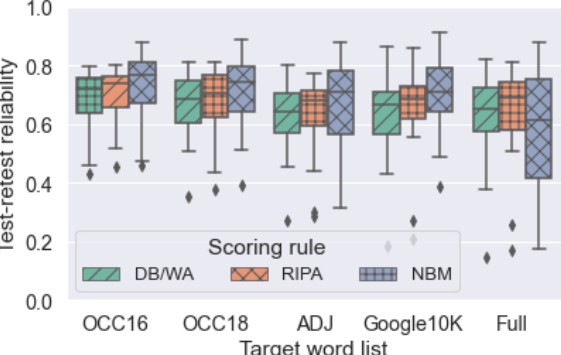

Figure 15: Test-retest reliability gender base pairs. The word embeddings are trained with GloVe on WikiText-103.

 **(b) Sexual orientation base pairs**

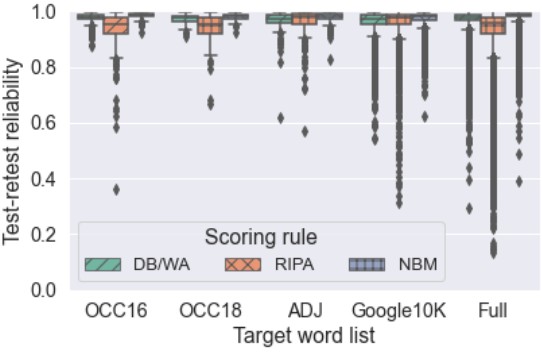

Figure 16: Test-retest reliability of target words (bias scores averaged across sexual orientation base pairs). The word embeddings are trained with SGNS on r/AskHistorians.

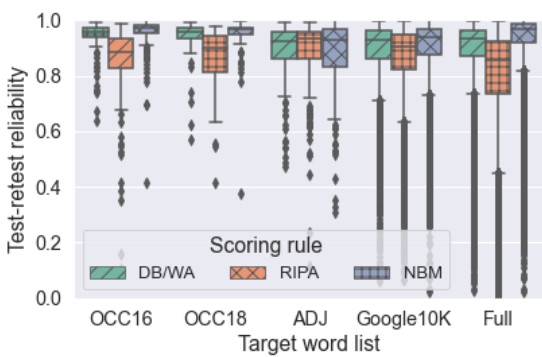

Figure 17: Test-retest reliability of target words (bias scores averaged across sexual orientation base pairs). The word embeddings are trained with SGNS on r/AskScience.

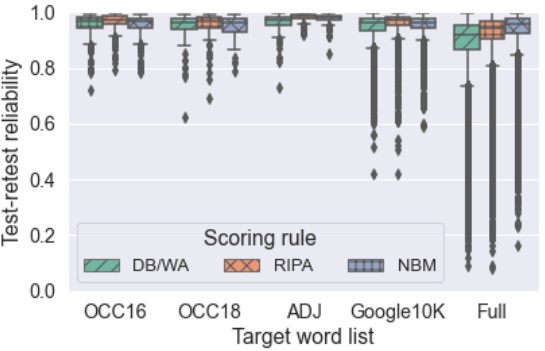

Figure 18: Test-retest reliability of target words (bias scores averaged across sexual orientation base pairs). The word embeddings are trained with SGNS on WikiText-103.

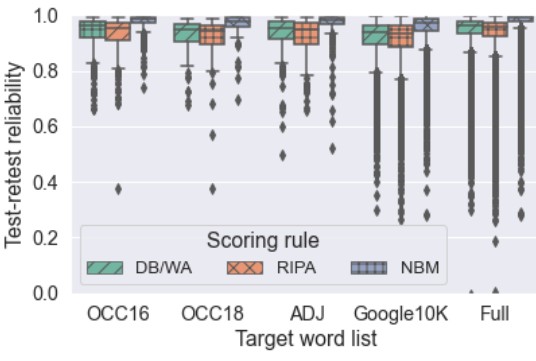

Figure 19: Test-retest reliability of target words (bias scores averaged across sexual orientation base pairs). The word embeddings are trained with GloVe on r/AskHistorians.

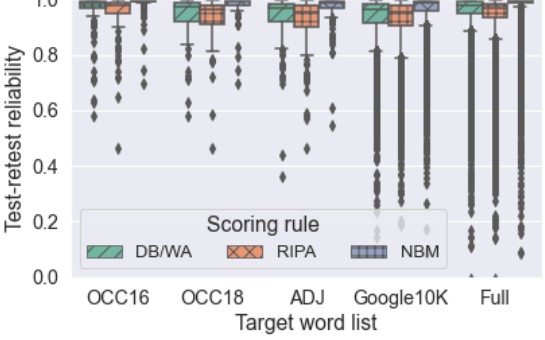

Figure 20: Test-retest reliability of target words (bias scores averaged across sexual orientation base pairs). The word embeddings are trained with GloVe on r/AskScience.

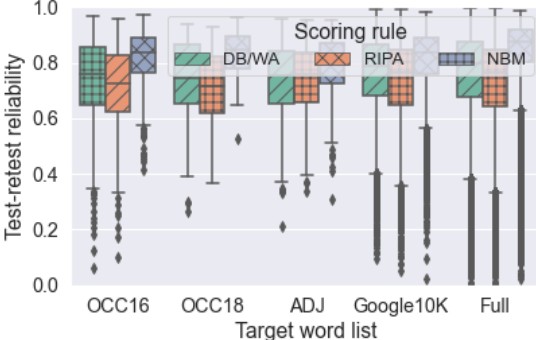

Figure 21: Test-retest reliability of target words (bias scores averaged across sexual orientation base pairs). The word embeddings are trained with GloVe on WikiText-103.

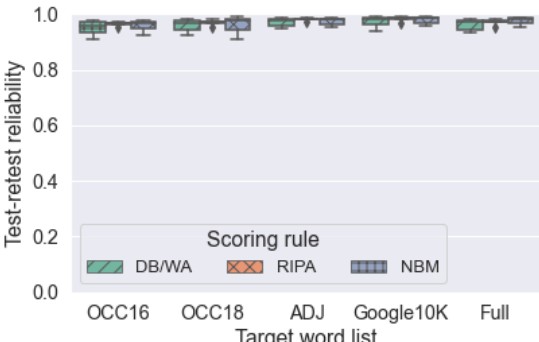

Figure 22: Test-retest reliability of sexual orientation base pairs. The word embeddings are trained with SGNS on r/AskHistorians.

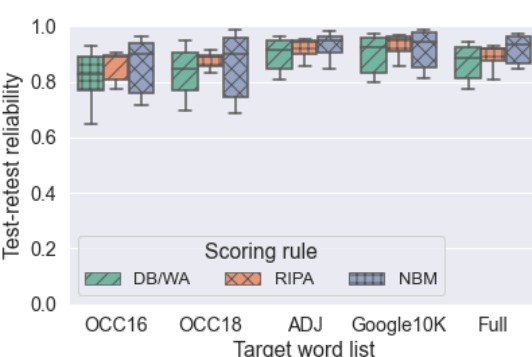

Figure 23: Test-retest reliability of sexual orientation base pairs. The word embeddings are trained with SGNS on r/AskScience.

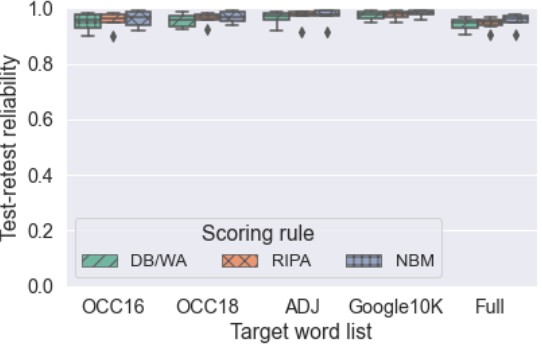

Figure 24: Test-retest reliability of sexual orientation base pairs. The word embeddings are trained with SGNS on WikiText-103.

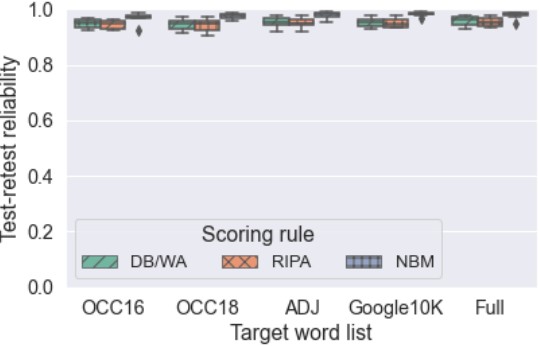

Figure 25: Test-retest reliability of sexual orientation base pairs. The word embeddings are trained with GloVe on r/AskHistorians.

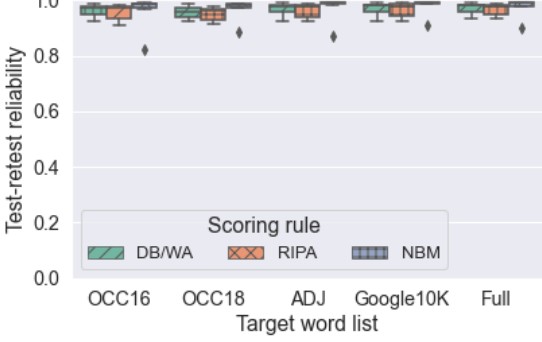

Figure 26: Test-retest reliability of sexual orientation base pairs. The word embeddings are trained with GloVe on r/AskScience.

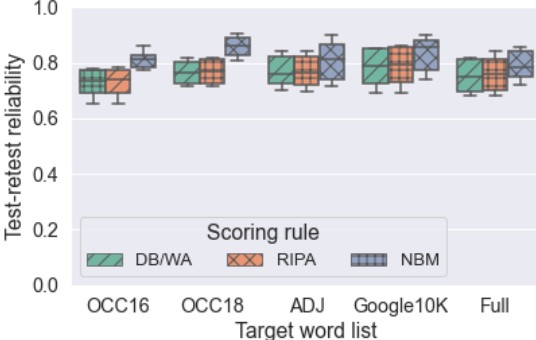

Figure 27: Test-retest reliability of sexual orientation base pairs. The word embeddings are trained with GloVe on WikiText-103.

 **Inter-Rater Consistency**

 **(a) Gender base pairs**

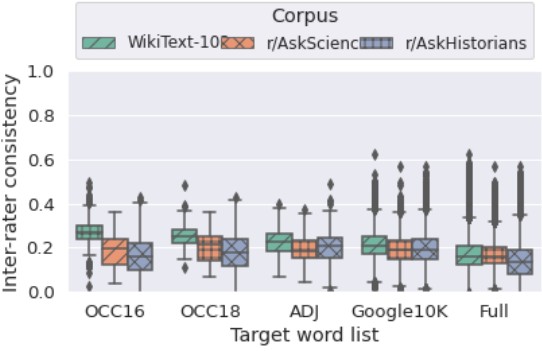

Figure 28: Inter-rater consistency of target words (bias scores averaged across gender base pairs). The word embeddings are trained with SGNS.

Figure 29: Inter-rater consistency of gender base pairs. The word embeddings are trained with SGNS.

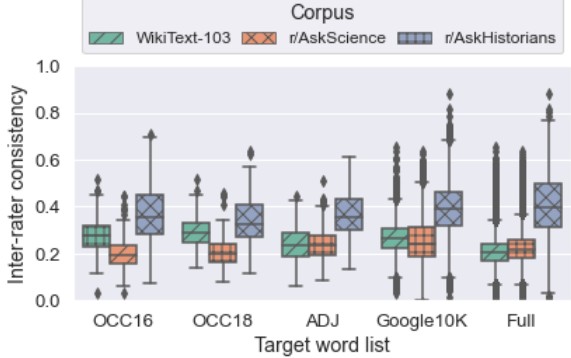

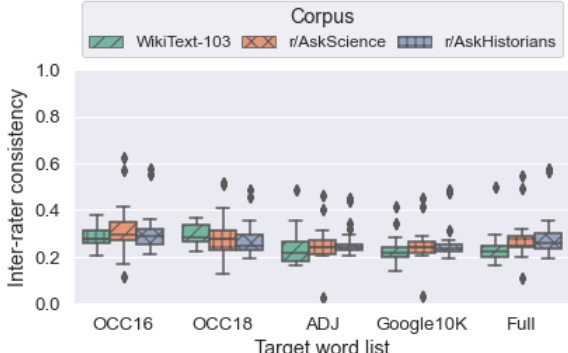

Figure 30: Inter-rater consistency of target words (bias scores averaged across gender base pairs). The word embeddings are trained with GloVe.

Figure 31: Inter-rater consistency of gender base pairs. The word embeddings are trained with GloVe.

348 **(b) Sexual orientation base pairs**

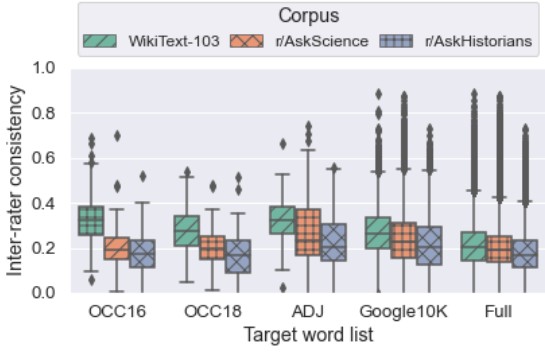

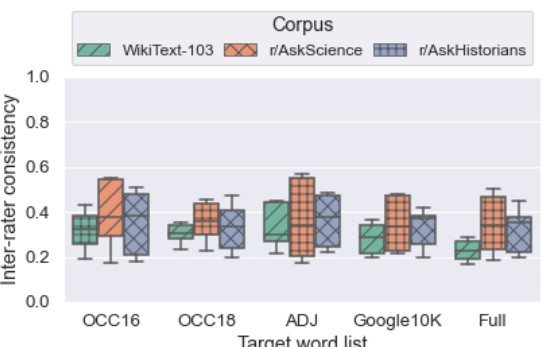

Figure 32: Inter-rater consistency of target words (bias scores averaged across sexual orientation base pairs). The word embeddings are trained with SGNS.

Figure 33: Inter-rater consistency of sexual orientation base pairs. The word embeddings are trained with SGNS.

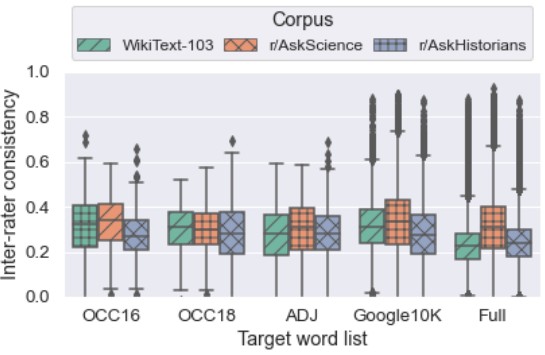

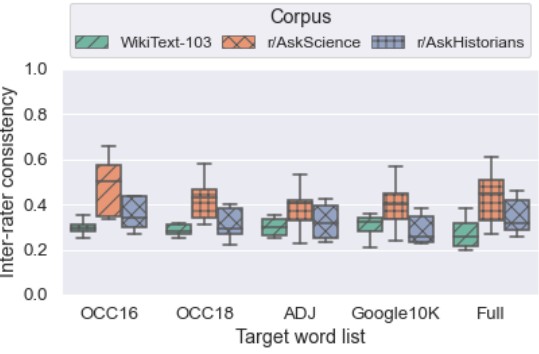

Figure 34: Inter-rater consistency of target words (bias scores averaged across sexual orientation base pairs). The word embeddings are trained with GloVe.

Figure 35: Inter-rater consistency of sexual orientation base pairs. The word embeddings are trained with GloVe.

349 **(c) Pearson correlation coefficients**

|  | SGNS | Glove |
| --- | --- | --- |
| RIPA & NBM | 0.689 | 0.816 |
| DB/WA & RIPA | 0.781 | 0.989 |
| DB/WA & NBM | 0.844 | 0.810 |

Table 4: Pearson's $r$ scores for bias scores of all corpora averaged over the gender base pairs. All with significance $p < 0.05$.

|  | SGNS | Glove |
|---|---|---|
| RIPA & NBM | 0.723 | 0.763 |
| DB/WA & RIPA | 0.884 | 0.986 |
| DB/WA & NBM | 0.654 | 0.748 |

Table 5: Pearson's $r$ scores for bias scores of all corpora averaged over the sexual orientation base pairs. All with significance $p < 0.05$.

# Internal Consistency

## (a) Gender base pairs

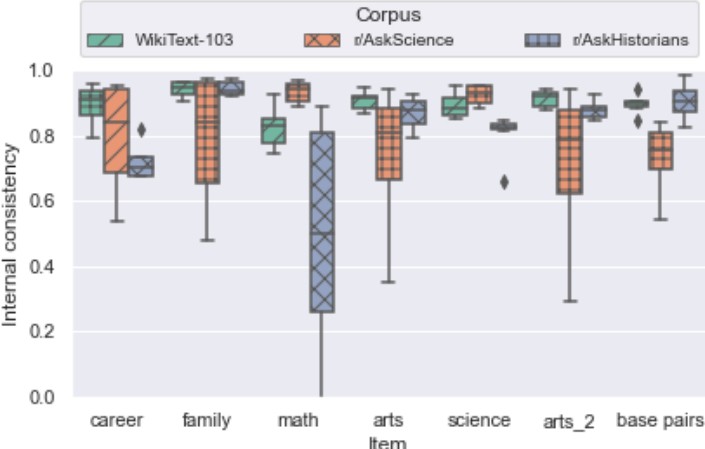

Figure 36: Distribution of internal consistency scores of gender related queries (e.g., career) and the ensemble of gender base pairs (base pairs) across corpora.

## (b) Sexual orientation base pairs

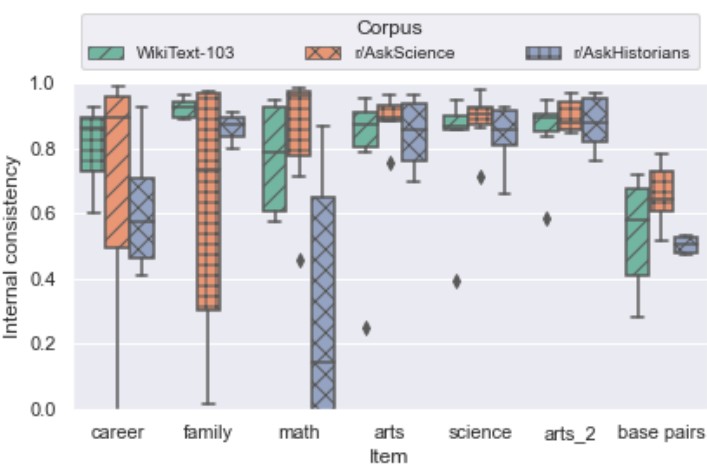

Figure 37: Distribution of internal consistency scores of the same gender related queries (e.g., career) which are also related to sexual orientation and the ensemble of sexual orientation base pairs (base pairs) across corpora.

**Factors influencing the reliability of gender or sexual orientation base pairs and target words**

**(a) Gender base pairs**

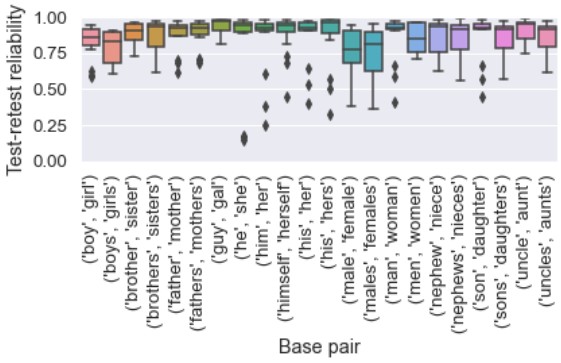

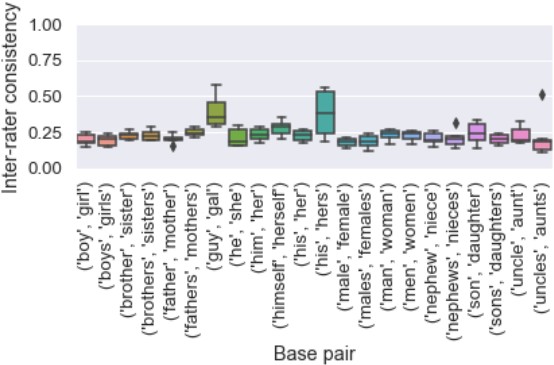

Figure 38: Test-retest reliability of different gender base pairs on full vocabularies.

Figure 39: Inter-Rater consistency of different gender base pairs on full vocabularies.

**(b) Sexual orientation base pairs**

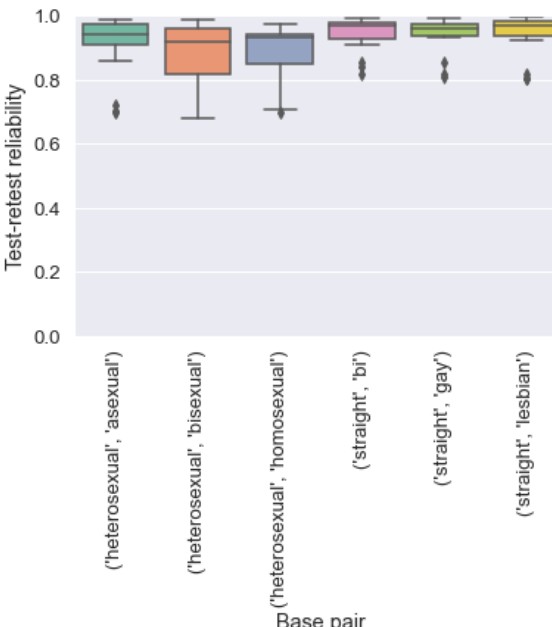
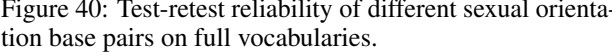

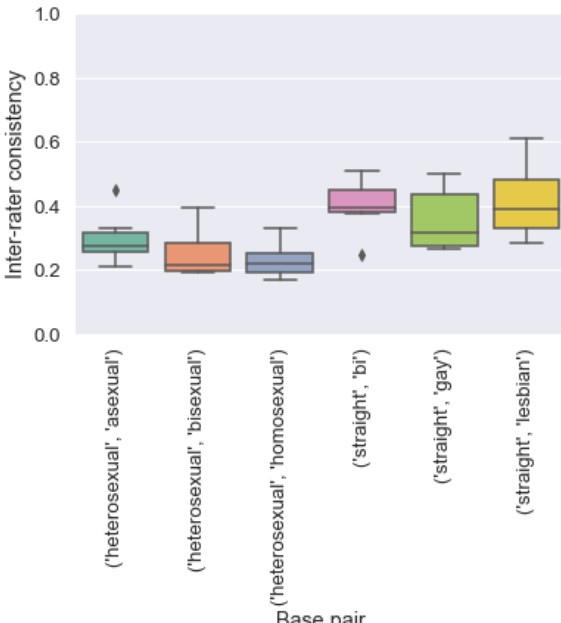

Figure 40: Test-retest reliability of different sexual orientation base pairs on full vocabularies.

Figure 41: Inter-Rater consistency of different sexual orientation base pairs on full vocabularies.

