# OpenReview forum: "[Re] Assessing the Reliability of Word Embedding Gender Bias Measures"
_ML_Reproducibility_Challenge/2021/Fall — Reject_

### Official Review · Reviewer_UetD · 2022-02-28
**Review of [Re] Assessing the Reliability of Word Embedding Gender Bias Measures**

**Rating:** 8
**Confidence:** 4

**Review:**

Overall, the paper was well written and easy to follow, along with a complete review of the reproducibility of results from the original paper.

Pros
+ The authors of this reproducibility paper covers writing the reproducibility summary, scope, discussion on result and recommendations for the reproducibility, and results beyond the paper (providing that the random seeds count should be higher).
+ The paper also gives a brief summary of the paper and description of the models / experiments for the readers to follow up on.

Cons
- More information on how the authors achieved that the random seeds count was the only reason for the non-reproducibility would have been helpful. Did the authors try a higher number that reached more stable results? Did the original paper mention that the random seed may cause some variability in the results?

Questions:
- For the GloVe embedding model, was the link to the GitHub provided by the authors? The code might not have been provided since it’s pretty well-known work.
- For the Test-Retest reliability test, what were the actual numeric delta between the scores? Was it significant?
- Did the paper provide which 32 random seeds were used to reproduce the results?

Nits:
- Line 91: First sentence sounds like a command. Starting with something like "You can download the ...."  would be better.
- Words 'Figure' and 'Table' should be capitalized
- Figure 1: better subcaptions (to denote which ones are the reproduced results v results from the paper would be helpful, not just in the caption)
- Figure 1: what are DB/WA, R1PA, NBM?

---

### Official Review · Reviewer_ggzg · 2022-03-08

**Rating:** 7
**Confidence:** 4

**Review:**

The review is clean and the authors use different random seeds. it is successfully re-implemented.

---

### Meta-Review · Area_Chair_RAKH · 2022-04-07

**Recommendation:** Reject
**Confidence:** 5

**Metareview:**

While the paper has good reviews, it lacks the standard for acceptance to the Reproducibility Challenge for a couple of reasons. The authors used readily available code, so it is expected the authors would perform an extensive and extra hyper-parameter search. However, it was not evident from the report. Secondly, having access to the code also calls for more ablation studies. While the authors do investigate different word lists, that itself isn't enough to justify the level of ablation expected from such a study. Having said that, the report itself is well formatted and should be an useful datapoint on the replicability of the original codebase.

---

### Decision · Program_Chairs · 2022-04-09

Reject